# Antibody–Drug Conjugates for the Treatment of Breast Cancer

**DOI:** 10.3390/cancers13122898

**Published:** 2021-06-09

**Authors:** Chiara Corti, Federica Giugliano, Eleonora Nicolò, Liliana Ascione, Giuseppe Curigliano

**Affiliations:** 1Division of New Drugs and Early Drug Development for Innovative Therapies, European Institute of Oncology, IRCCS, Via Ripamonti 435, 20141 Milan, Italy; chiara.corti@ieo.it (C.C.); federica.giugliano@ieo.it (F.G.); eleonora.nicolo@ieo.it (E.N.); liliana.ascione@ieo.it (L.A.); 2Department of Oncology and Haematology (DIPO), University of Milan, Via Festa del Perdono 7, 20122 Milan, Italy

**Keywords:** breast cancer, target therapy, antibody, antibody–drug conjugates, ADCs

## Abstract

**Simple Summary:**

Metastatic breast cancer (BC) is currently an incurable disease. Besides endocrine therapy and targeted agents, chemotherapy is often used in the treatment of this disease. However, lack of tumor specificity and toxicity associated with dose exposure limit the manageability of cytotoxic agents. Antibody–drug conjugates (ADCs) are a novel and evolving class of antineoplastic agents. By merging the selectivity of monoclonal antibodies with the cytotoxic properties of chemotherapy, researchers aim to optimize the therapeutic index of anticancer drugs. Some of these compounds, such as trastuzumab deruxtecan, showed activity not only in HER2-positive, but also in HER2-low BC patients, possibly due to the bystander effect. In this review, the current clinical landscape about ADC development for BC treatment will be discussed, as well as the possible limitations of this treatment class.

**Abstract:**

Metastatic breast cancer (BC) is currently an incurable disease. Besides endocrine therapy and targeted agents, chemotherapy is often used in the treatment of this disease. However, lack of tumor specificity and toxicity associated with dose exposure limit the manageability of cytotoxic agents. Antibody–drug conjugates (ADCs) are a relatively new class of anticancer drugs. By merging the selectivity of monoclonal antibodies with the cytotoxic properties of chemotherapy, they improve the therapeutic index of antineoplastic agents. Three core components characterize ADCs: the antibody, directed to a target antigen; the payload, typically a cytotoxic agent; a linker, connecting the antibody to the payload. The most studied target antigen is HER2 with some agents, such as trastuzumab deruxtecan, showing activity not only in HER2-positive, but also in HER2-low BC patients, possibly due to a bystander effect. This property to provide a cytotoxic impact also against off-target cancer cells may overcome the intratumoral heterogeneity of some target antigens. Other cancer-associated antigens represent a strategy for the development of ADCs against triple-negative BC, as shown by the recent approval of sacituzumab govitecan. In this review, we discuss the current landscape of ADC development for the treatment of BC, as well as the possible limitations of this treatment.

## 1. Introduction

Metastatic breast cancer (BC) remains an incurable disease [1]. Besides endocrine therapy and targeted agents, chemotherapy is often used in the treatment of this disease [2]. However, lack of tumor specificity and toxicity associated with dose exposure limit the manageability of cytotoxic agents.

Antibody–drug conjugates (ADCs) represent a relatively new and evolving class of anticancer drugs [3]. By merging the target specificity of monoclonal antibodies (mAbs) with the cancer-killing abilities of cytotoxic drugs, chemotherapy is thought to be mostly delivered to cells that express a selected target antigen, therefore improving the therapeutic index [3].

Historically, the association of cytotoxic agents with Ab species dates back to 1950s [4]. The subsequent two decades paved the way to the production of mAbs with improved targeting accuracy, including ADCs, which showed promising results in both in vitro and in vivo models of cancer [3]. In the 1980s, the first clinical trials of ADCs for the treatment of cancer patients revealed worrying drug toxicities, without a clear clinical benefit [5]. Fortunately, this trend ended with the first Food and Drug Administration (FDA) approvals for ADCs in the 2000s [6].

In this context, BC played a prominent role in ADC evolution, since three out of the nine ADCs that are currently FDA-approved are for the treatment of BC (Figure 1 and Figure 2). In particular, ado-trastuzumab emtansine (T-DM1), an ADC targeting the human epidermal growth factor receptor 2 (HER2), was the first compound approved for the treatment of trastuzumab-resistant metastatic BC in 2013 [7].

In this light, the aim of this review is to provide a comprehensive overview of the mechanism, safety and efficacy of approved and investigational ADCs for BC.

## 2. Structure of Antibody–Drug Conjugates

ADCs consist of three core components: the antibody, directed to a target antigen; the payload, typically a cytotoxic agent; a linker, which connects the antibody to the payload. The spectrum of variation of each component in different ADCs strongly influences their pharmacological and clinical properties (Figure 2).

### 2.1. Target-Directed Antibody

ADCs contain a chimeric or humanized Abs backbone, which reduces both acute hypersensitivity reactions and the generation of neutralizing anti-drug Abs [8]. In particular, immunoglobulin G (IgG) represents the main Ab backbone in ADCs [9]. Specifically, IgG1 isotype architecture not only is easier to produce, but shows excellent complement-fixation and FcγR-binding capacities, supporting both Ab-dependent cellular cytotoxicity (ADCC) and complement-dependent cytotoxicity (CDC) reactions [9,10].

The ideal mAb target should be a cell surface protein that is exclusively expressed on tumor cells, in order to limit the risk of systemic toxicity. In this regard, antigens expressed on solid tumors are often expressed on normal cells as well [3]. Therefore, such antigens are defined as “tumor-associated” rather than “tumor-specific” [11]. Consequently, for all these compounds, toxicity may occur according to the spectrum of expression of the specific target by normal cells (on target—off tumor toxicity) and cancer cells (on target—on tumor toxicity) [11,12]. Note that toxicity has also been reported regardless of the specific target distribution (off target—off tumor toxicity) (Figure 3).

As for BC, examples of successful targets include HER2 and Trophoblast cell surface antigen 2 (TROP2) [13]. In this regard, BCs with high levels of intratumor or intertumor heterogeneity in HER2 expression respond less favorably to T-DM1 compared with those with homogeneous HER2 expression [14,15].

Activity of ADCs is influenced not only by target expression, but also by its turnover, internalization, lysosomal processing and degradation [3]. Moreover, if targets are functionally oncogenic, they tend to be resistant to downregulation, a mechanism of drug resistance. In this view, oncogenic targets can be exploited for additional ADC activity via Ab-mediated suppression of downstream oncogenic signaling pathways [3].

### 2.2. The Linker

The linker is a key component of ADCs because it affects the pharmacokinetics (PK) aspects, such as drug stability into the bloodstream, tumor cell permeability, the number of payload molecules carried by each Ab (i.e., drug-to-Ab ratio, DAR) and the extent of the bystander effect. The main role of a linker is to connect the Ab to the payload, while the drug circulates in the bloodstream [16]. If linkers are unstable in plasma, the payload may be released prematurely, with consequent challenging systemic toxicity and reduced payload delivery at the tumor site [17]. Of note, numerous ADCs carry potent cytotoxic warheads, with toxicity profiles unsuitable for systemic delivery [3]. Another important role of the linker is to ensure an adequate release of the payload within tumor cells [3,18].

Linkers typically consist of two classes: cleavable and non-cleavable [3]. Cleavable linkers break down in cancer cells and release the payload in response to environmental factors, such as pH (acid-labile linkers), reduction-oxidation conditions (disulfide linkers) or proteolytic enzymes (protease-cleavable linkers). Although cleavable linkers show overall stability into the bloodstream, they may decay in plasma over time [17]. Conversely, non-cleavable linkers consist of stable bonds in plasma and resistance to proteolytic degradation, so that the payload is released upon cleavage relying on lysosomal degradation of the entire Ab–linker complex [9,17]. Lower membrane permeability may affect these type of linkers [9].

### 2.3. The Payload

Since ADCs aim at improving the therapeutic index of antineoplastic agents, recent experimentations were conducted with highly potent chemotherapy drugs, which may be cytotoxic at sub-nanomolar concentrations and would have an unfavorable toxicity profile if administered systemically [19].

The DAR is used to express the average amount of payload moieties linked to each mAb. In general, if similar PK profiles are hypothesized, ADCs with high DARs are supposed to be more potent in vitro [3]. However, some ADCs might be rapidly cleared from the bloodstream by the liver, resulting in similar activity to ADCs with lower DARs in preclinical models [20,21]. Some payloads are also able to account for the “bystander effect”: the diffusion of cell-permeable (hydrophobic) payloads from within cells harboring the target antigen into surrounding cells, on which the drug can exert a cytotoxic effect, irrespective of the target antigen expression [3,22].

DM1, a synthetic derivative of maytansine, acts by inhibiting microtubule polymerization and is the payload conveyed by T-DM1, the first ADC approved for BC [7,23]. Sacituzumab govitecan (SG) is a newly approved ADC coupling an anti-TROP2 Ab with SN-38, the active metabolite of irinotecan [24]. Finally, deruxtecan (DX-d), an anti-topoisomerase 1 (TOP1) compound, is the warhead of the anti-HER2 trastuzumab deruxtecan (T-DXd), which has been approved for BC and gastric cancer, while being under development for several tumor types [25].

## 3. Antibody–Drug Conjugates in HER2-Positive Breast Cancer

Since the identification of HER2 in 1982–1984, further findings led to the characterization of HER2-positive BC and to the pursuit of targeted agents [26]. HER2 is a transmembrane growth factor receptor, member of the HER protein family, along with HER1 (EGFR), HER3 and HER4 [9]. Several epidermal growth factor (EGF) ligands have been described, although none specific to HER2 [9]. The binding of a ligand to HER1, HER3 or HER4 causes receptor dimerization—with HER2 as the preferred dimer partner—followed by the activation of the tyrosine-kinase intracellular domain, with different downstream pathways being triggered [9]. Although many adult tissues express HER2 on the cell surface, the primary role of this protein has been associated to organogenesis [27]. As an oncogene, HER2 exerts its role especially due to gene overexpression, which boosts HER2 heterodimerization, inducing both cell transformation and tumorigenic growth [28,29,30]. HER2 somatic mutations have been described as well [31,32]. Mutations occur in the extracellular or kinase domain in about 90% of cases, while the transmembrane and juxtamembrane domains are mutated in ~7% and ~3% of cases, respectively [9]. Only in recent years their tumorigenic role, as well as their sensitivity to anti-HER2 agents has been elucidated [32].

As for BC, the oncoprotein HER2 is overexpressed in 25% of cases [33]. To date, different anti-HER2 compounds have proven to offer a significant clinical benefit in this disease, with ADCs representing the ultimate weapon of the therapeutic armamentarium [26,33]. Historically, early trials testing trastuzumab, the first anti-HER2 drug ever approved, demonstrated that tumor responses were restricted to patients whose tumors stained 3+ for HER2 on immunohistochemistry (IHC) or stained 2+ but had HER2 gene amplification (≥2 copies) on fluorescence in situ hybridization (FISH) [16,34,35]. From then on, both clinical trials and international guidelines tested and recommended anti-HER2 therapies, respectively [2,36,37].

In comparison with HER2-positive BCs, a greater proportion of patients (~45–50%) have BC classified as HER2-low, i.e., IHC of 1+ or 2+, with a negative FISH test [16,38]. For treatment decisions, HER2-low BCs have been considered HER2-negative so far, together with those displaying IHC 0+, and thus, they are not eligible for anti-HER2 therapies [2,16]. In a phase 2 study in a neoadjuvant setting, HER2 heterogeneity accounted for 10% of cases [39].

If, in the past, trastuzumab did not improve the outcomes of patients with HER2-low BC, novel and more potent anti-HER2 agents, such as ADCs, paved the way to new treatment options in the metastatic setting [40,41,42,43].

### 3.1. Ado-Trastuzumab Emtansine (T-DM1)

T-DM1 was the first ADC to be granted FDA approval [44]. It is an anti-HER2 ADC, which combines the anti-HER2 properties of trastuzumab with DM1, a derivative of the maytansinoid toxin, which inhibits tubulin polymerization [3]. The linker is a non-cleavable stable thioether and the DAR is about 3.5:1. PK studies showed that, while peak serum concentrations of total trastuzumab (conjugated plus naked antibodies) exceed those of the complete ADC by approximately 20%, concentrations of the DM1 payload are much lower [3,45]. Whereas the half-life of trastuzumab is 9–11 days, that of T-DM1 is approximately 4 days, maybe due to linker instability, Ab recycling or hepatic clearance of the ADC [3,45].

Preclinical data suggest that the antitumoral properties of T-DM1 reside (a) in the inhibition of HER2 signaling exerted by mAb trastuzumab, via blocking of ligand-independent HER2 dimerization; (b) in the ADCC induction by the IgG1 backbone; (c) in the internalization of the cytotoxic moiety by HER2 expressing cells [3,46]. The most frequent adverse events (AEs) are nausea, fatigue, thrombocytopenia, diarrhea and elevated levels of liver enzymes in up to 40% of patients [44]. Patients may also experience neuropathy, especially if exposure to the drug is prolonged (Figure 4) [44,47,48,49].

T-DM1 was initially approved by the FDA in 2013 for patients with HER2-positive metastatic BC previously treated with trastuzumab and a taxane, based on the results of the EMILIA trial [7]. This study demonstrated an improved progression-free survival (PFS) and overall survival (OS) in patients treated with T-DM1 compared to those treated with lapatinib and capecitabine.

In later lines, T-DM1 significantly improved both PFS and OS compared to treatment of physician’s choice (TPC) (TH3RESA) [50]. Recently, T-DM1 have reached the standard of care for patients with HER2-positive disease with residual disease at the time of surgery in the post-neoadjuvant setting (KATHERINE trial) [51].

As a first-line treatment, T-DM1 with or without pertuzumab was non-inferior in terms of PFS compared to trastuzumab plus a taxane for metastatic HER2-positive disease (MARIANNE trial) [52]. Because the addition of pertuzumab to trastuzumab and a taxane in this setting prolonged both PFS and OS, T-DM1 remains a second-line option [53,54]. In the neoadjuvant setting, docetaxel, carboplatin and trastuzumab plus pertuzumab increased the pathological complete response (pCR) rate in comparison with T-DM1 plus pertuzumab [55]. In conclusion, the development of T-DM1 as a first-line treatment for HER2-positive metastatic BC or as a neoadjuvant treatment has not been satisfactory thus far [47].

HER2-targeted combination treatment with T-DM1 is under investigation in ongoing clinical trials [26]: the CompassHER2-RD trial (NCT04457596) and the HER2CLIMB-02 trial (NCT03975647) are assessing the superiority of T-DM1 plus tucatinib versus T-DM1 alone in the adjuvant setting and in the metastatic setting, for patients previously treated with trastuzumab plus taxanes, respectively.

As for HER2-low metastatic BC, in two phase 2 trials testing the safety and efficacy of T-DM1 in patients pre-treated with at least trastuzumab, retrospective and exploratory analyses found poor clinical activity of T-DM1 [16]. However, T-DM1 was prospectively investigated in the same setting of HER2-positive but heterogeneous BC [16]. In a phase 2 study of neoadjuvant T-DM1 plus pertuzumab, HER2 heterogeneity was found in 10% of cases, with no patients with HER2 heterogeneity reaching a pCR, while 55% of patients harboring HER2-homogeneous tumors achieved pCR [16,39].

**Figure 4 cancers-13-02898-f004:**
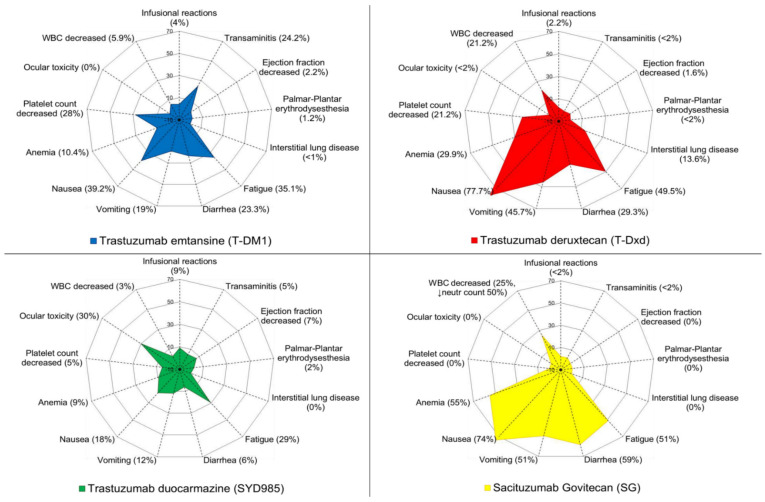
Toxicity profiles of the antibody–drug conjugates currently approved or in late stages of development. Abbreviations: WBC, white blood count; T-DM1, trastuzumab emtansine; T-Dxd, trastuzumab deruxtecan; neutr, neutrophil [7,25,42,56].

### 3.2. Fam-Trastuzumab Deruxtecan-Nxki (T-DXd)

Trastuzumab deruxtecan (DS-8201 or T-Dxd) is an ADC constituted of the anti-HER2 mAb trastuzumab and a cleavable tetrapeptide-based linker. The payload is a topoisomerase I inhibitor, an exatecan derivative, with a DAR of 8:1, thus allowing the delivery of a high concentration of the cytotoxic drug (Figure 2) [57]. T-Dxd is currently approved in United States and Japan for patients with advanced or metastatic HER2-positive BC after at least two prior anti-HER2-based regimens and is under accelerated assessment in Europe (Figure 1) [47].

This ADC is internalized by target cells and trafficked to lysosomes, after binding to HER2 [3]. The linker then undergoes cleavage by lysosomal cathepsins, which are upregulated in tumor cells [58,59]. The blocking of the HER2 signaling via inhibition of the ligand-independent HER2 dimerization by the anti-HER2 mAb moiety occurs with T-DXd according to preclinical studies [3].

In 2015, a phase I, first-in-human study enrolled pretreated patients with advanced HER2-positive BC, or with gastric or gastro-esophageal neoplasms, to investigate safety and activity profiles of T-DXd [57]. Patients enrolled had received a median of seven previous therapies and 70% of BCs were hormone receptor (HR)-positive tumors [47]. Median duration of treatment was 8.3 months at the data cutoff and 52% of patients had discontinued therapy, mostly due to progression of disease [57,60]. The objective response rate (ORR) was 59.5%, with a median time to response (TTR) of 1.6 months and median duration of response (DoR) of 20.7 months [47,60]. Interestingly, ORR was independent of previous pertuzumab treatment and HR status [47,60]. Median PFS was 22.1 months and median OS was not reached [47].

In 2019, the DESTINY-Breast01 clinical trial, a two-part, open-label, single-group, multicenter, phase II study, investigated T-DXd in HER2-positive metastatic BC patients (N = 184) heavily pretreated and who previously received T-DM1 [25]. ORR was 60.6% after a median follow-up of 11.1 months, with median DoR of 14.8 months and median PFS of 16.4 months [25]. This trial led to accelerated FDA approval of T-DXd for patients with unresectable or metastatic HER2-positive BC who received two or more anti-HER2 therapies in the metastatic setting [61].

Interestingly, T-DXd seemed to be active in patients with brain metastases, with an ORR of 58.3% and a median PFS of 18.1 months, similarly to the overall study population [9,62]. However, the study protocol excluded patients with untreated or progressing brain metastases, so it is still unknown whether T-DXd is effective against a central nervous system (CNS) disease [47]. In this regard, a phase II study is currently investigating T-DXd in patients with both HER2-positive and HER2-low BC with untreated or progressive brain metastases (DEBBRAH trial, NCT04420598).

In both early and advanced setting of HER2-positive BCs, ongoing randomized phase III clinical trials aim at investigating the clinical benefit of T-DXd compared with other approved anti-HER2 treatments, such as lapatinib plus capecitabine (NCT03523585) and T-DM1 (NCT03529110) [16,47]. Finally, a phase III, multicenter, randomized, open-label study is evaluating the invasive disease-free survival (IDFS) of T-DXd versus T-DM1 in patients with high-risk HER2-positive BC presenting with residual invasive disease in breast or axillary lymph nodes in the post-neoadjuvant setting (DESTINY-Breast05, NCT04622319) [16,47].

As for T-DXd activity, both specific features of the linker and the membrane-permeable nature of the payload may account for the bystander killing effect, which is the potential to also provide a cytotoxic impact against off-target cancer cells, due to diffusion of the free cytotoxic moiety from the antigen-positive dying cells [16,62]. The bystander effect might induce clinical activity even in HER2-low or HER2-heterogenous BCs, while ensuring a safe therapeutic index [47].

In this regard, several studies have focused on T-DXd in patients with HER2-low tumors [9,59]. In the phase Ib study assessing the activity of T-DXd in patients with HER2-low metastatic BC, patients mostly had HR-positive disease (87%), harbored visceral disease at baseline and were heavily pretreated (median 7.5 prior therapies) [63]. The ORR was 37%, with a median DoR of 10.4 months, median PFS of 11.1 months and median OS of 29.4 months [63]. Patients with HER2 IHC 1+ or 2+ did not show relevant differences in ORR (HER2 1+, 35.7% vs. HER2 2+, 38.5%) [63]. HR-negative tumors showed lower ORR (14.4%) in comparison with HR-positive tumors (40.3%) [63].

Subsequently, two randomized, phase III trials compared T-DXd versus TPC in HER2-low, unresectable or metastatic BC (NCT03734029/DESTINY-Breast04 and NCT04494425/DESTINY-Breast06, in late lines post-chemotherapy and chemo-naïve post-endocrine therapy, respectively) [16]. While DESTINY-Breast04 permits inclusion of any HER2-low BC, irrespective of HR status, DESTINY-Breast06 limits the inclusion of only HR-positive tumors.

The toxicity profile of T-DXd is not trivial, even though no severe AEs raised issues of concern about its safe use in the clinical setting (Figure 4). The most common any-grade AEs are gastrointestinal (nausea, 78%; vomiting, 46%; constipation, 36%; decreased appetite, 31%; diarrhea, 29%; abdominal pain, 17%) and hematological [3,25]. The most common grade ≥3 toxicities were neutropenia (20.7%), anemia (8.7%), nausea (7.6%), leukopenia (6.5%), lymphopenia (6.5%) and fatigue (6.0%) [25]. Of note, interstitial lung disease (ILD), pneumonitis or organizing pneumonia represented concerning issues during early-phase clinical trials, as a significant prevalence of these AEs, as well as treatment-related deaths, were reported [47,60]. In DESTINY-Breast01, all-grade drug-related ILD occurred in 25 patients (total, 13.6%; grade 1–3, 10.9%; grade 3–4, 0.5%; grade 5, 2.2%) [25]. Although the update analysis of the trial confirmed the benefit of the drug (PFS = 19.4 months), safety data showed a higher incidence of ILD (15.2%) [64]. In this regard, a post hoc analysis of pooled data from patients included in two clinical trials with advanced solid tumors and BC (NCT02564900 and DESTINY-Breast01, respectively) confirmed a higher incidence of any-grade ILD for BC patients in comparison with the overall study population (18.1% for BC population versus 16.8% overall), with an earlier median time to onset (134 days for BC patients versus 208 days in the overall population) [47]. Should ILD be suspected, consultation of a pneumologist, high-resolution computed tomography, pulmonary function tests and oxygen saturation are mandatory [47]. At present, in case of grade 1 ILD, experts recommend holding the subsequent T-DXd dose and administering systemic steroids (0.5–1 mg/kg of prednisone or equivalent dose). Treatment should be resumed only upon resolution of the AE [47]. Permanent discontinuation of T-DXd is suggested for grade ≥2 ILD with prompt start of steroids [65]. Additionally, also in light of the recent coronavirus disease 2019 (COVID-19), caused by severe acute respiratory syndrome coronavirus 2 (SARS-CoV-2), other etiologies such as respiratory infection should be considered and excluded [66].

As cardiomyocytes rely on HER2 growth signaling to maintain their homeostasis, cardiotoxicity may follow upon blockade of this pathway, in the form of reduced left ventricular ejection fraction (LVEF) [16]. Unlike this well-known cardiotoxicity profile of trastuzumab, only three patients displayed a decrease in LVEF with T-DXd and all of them fully recovered after drug interruption [25]. Any-grade QT interval prolongation was described for nine patients (4.9%) and of grade 3 for two patients (1.1%) [25].

As for future perspectives, several clinical trials are evaluating combination treatment involving T-DXd and aromatase inhibitors (NCT04553770), chemotherapy (DESTINY-Breast08, NCT04556773), tucatinib (HER2CLIMB-04, NCT04539938) and immune checkpoint inhibitors (ICIs) (BEGONIA trial, NCT03742102; KEYNOTE-797, NCT04042701; DESTINY-Breast07, NCT04538742; DESTINY-Breast08, NCT04556773; NCT03523572) (Table 1).

### 3.3. Trastuzumab–Duocarmazine (SYD985)

Trastuzumab duocarmazine is an ADC constituted of a humanized anti-HER2 mAb (trastuzumab) linked to a duocarmycin warhead via a cleavable linker (DAR, 2.8:1) [47]. In fact, the cytotoxic moiety is a cell-permeable pro-drug (seco-duocarmycin-hydroxybenzamide-azaindole, seco-DUBA) that is cleaved into the active toxin (DUBA) in intracellular lysosomes by proteases, after internalization [16]. The payload then alkylates the DNA, causing DNA damage and cell death, also through the bystander killing effect, possibly due to interstitial cleavage of trastuzumab–duocarmazine by malignant cells secreting cathepsin B, with the generation of free DUBA [16,62,67,68,69]. Despite its DAR being lower than T-DM1, trastuzumab–duocarmazine displays higher activity than T-DM1 in HER2-low, patient-derived xenograft BC models [16,70].

In the phase I study conducted in patients with treatment-refractory locally advanced or metastatic solid tumor and variable HER2 status, 47 HER2-low BC patients were enrolled in the BC dose expansion cohort [16]. The experimental ADC showed an encouraging ORR of 33% in HER2-positive BC and of 28% in HR-positive HER2-low BC and 40% for HR-negative HER2-low BC [16,42].

As for the safety profile, the most frequent treatment-related AEs are fatigue (33%), conjunctivitis (31%) and dry eye (31%, with 7% of patients with grade 3 events) [42]. It is worth noticing that ocular toxicity dominates the safety profile of SYD985 (Figure 3 and Figure 4) [42]. Besides the aforementioned most frequent ocular AEs, keratitis and blurred vision were reported as well [42]. Four patients (3%) had grade 3 conjunctivitis in the dose-expansion cohort [42]. Dose reduction, decrease in the administration frequency, or prophylactic use of eye drops did not appear to impact these AEs [42]. Nevertheless, most patients were able to continue trastuzumab–duocarmazine and most ocular problems were reported to improve, therefore suggesting that these AEs are manageable [16,42]. LVEF decrease was detected in the phase I study of trastuzumab-duocarmazine: 10 patients (7%) with grade 1–2 and 1 patient (3%) with grade 3. In eight patients (5%), an absolute decrease of at least 10% from baseline to a value below 50% was reported [42].

Trastuzumab duocarmazine is also being investigated in a phase III study (TULIP trial), in which it is compared with TPC in patients with HER2-positive metastatic BC after at least two anti-HER2 regimens for advanced disease or in patients progressed on T-DM1 treatment (TULIP trial, NCT03262935).

### 3.4. Disitamab Vedotin (RC48-ADC)

Disitamab vedotin is an ADC composed of an anti-HER2 humanized mAb (disitamab) coupled to four molecules (DAR, 4:1) of monomethyl auristatin E (MMAE) warhead, by means of a protease cleavable linker [71]. In the phase I study conducted in patients with HER2-positive metastatic BC, safety aspects included grade 3 neutropenia (10%), leukopenia (6.7%) and transaminase elevation (3.3%), with no grade 4 AEs (Figure 3) [47,72]. Presently, a phase II study is elucidating the efficacy of disitamab vedotin at a dose of 2 mg/kg every 2 weeks versus capecitabine plus lapatinib in HER2-positive metastatic BC (NCT03500380). Finally, a phase III randomized trial aims at assessing the efficacy of RC48-ADC in comparison with TPC in patients with HER2-low metastatic BC, who progressed on or after the first treatment line (NCT04400695).

### 3.5. XMT-1522 (TAK-522)

XMT-1522 is an ADC that targets HER2 through mAb HT-19, which binds to a different HER2 epitope than trastuzumab [47]. HT-19 is linked to the payload, an auristatin derivative, by a degradable cysteine-linkage with a DAR of 12:1 [16]. XMT-1522 seems to trigger a bystander killing effect and to be more active than T-DM1 in HER2-positive and HER2-low patient-derived xenograft and in early phase clinical trials, although the development of the drug has been put on a partial clinical hold by the FDA due to safety issues [73,74].

### 3.6. MM-302

The ADC MM-302 mediates a HER2-targeted release of high doses of anthracycline, while reducing exposure to healthy tissues, such as cardiomyocytes [13,75]. The binding site of the drug is a different HER2 domain than the one exploited by trastuzumab [76]. Of note, a synergistic action with the latter is suggested by preclinical evidence [76]. Even though a phase I trial confirmed the activity of this ADC in pretreated metastatic BC, regrettably, the phase 2 HERMIONE trial was closed due to a lack of significant activity of MM-302 combined to trastuzumab when compared with TPC [13,77,78].

### 3.7. MEDI-4276

This drug is a bispecific ADC, which targets two distinct epitopes of HER2 [16]. The bispecific nature may increase internalization with consequent higher payload release and enhanced killing of cancer cells [13,79]. In MEDI-4276, the mAb is coupled with a microtubule inhibitor (AZ13599185) through a cleavable linker (DAR, 4:1) [80]. On the basis of the activity observed in preclinical models, a phase 1 dose-escalation clinical trial is elucidating the role of this ADC in HER2-positive metastatic BC and gastric cancer [81].

Notably, several other anti-HER2 ADCs are currently being investigated, including A166, ARX788, BAT8001 and PF-06804103, as summarized in Table 1 [9].

## 4. Other Targets Exploited by Antibody–Drug Conjugates in Breast Cancer

### 4.1. Trop2

Trop2 is encoded by the *TACSTD2* gene and is a transmembrane glycoprotein that serves as intracellular calcium signal transducer [9,82]. Trop2 is expressed in many normal tissues, such as uterus, skin, esophagus, bladder, oral mucosa, nasopharynx and lungs [9]. Trop2 is overexpressed in several epithelial tumors, including urothelial, breast, gynecological, lung and gastro-intestinal carcinomas and is associated to poor outcomes [9,83]. Although the exact role of Trop2 in cell signaling is yet to be elucidated, the main pathways associated to Trop2 are extracellular signal-regulated kinases/mitogen-activated protein kinases (ERK/MAPK) and the nuclear factor kappa-light-chain-enhancer of activated B cells (NF-kB) [84].

#### 4.1.1. Sacituzumab Govitecan (IMMU-132)

By means of a cleavable linker, the ADC SG conjugates a humanized anti-Trop-2 mAb hRS7 IgG1k with the cytotoxic warhead SN-28, the active metabolite of the topoisomerase I inhibitor irinotecan (DAR, 7.6–8:1) (Figure 2) [47,85]. After antigen binding, the mAb, in free or conjugated form, is internalized into target cells, then trafficked to lysosomes [86]. The low pH found in lysosomes facilitates the hydrolysis of the linker and the consequent release of the payload [47]. Due to the membrane-permeable nature of SN-38, some drug molecules are also released in the tumor microenvironment, contributing to killing neighboring tumor cells (bystander killing effect) [9,87].

The first phase I trial of SG enrolled 25 patients with different tumor histologies, including four patients with triple-negative breast cancer (TNBC) [85].

Besides being expressed in TNBC, Trop-2 is also frequently expressed in HR-positive BC [9]. Consequently, SG activity has been investigated also in this BC subtype, with encouraging outcomes coming from the HR+/HER2- metastatic BC cohort of the IMMU-132-01 (NCT01631552) basket trial [9]. In this phase I/II multicenter trial, patients with different solid neoplasms who have received at least one previous therapy in the metastatic setting were eligible for enrolment [56]. Patients with active brain metastases or under high-dose steroid treatment for at least 4 weeks before enrolment were not eligible as per the study protocol [56]. In 2019, data about patients with metastatic TNBC were published [88]. The ORR was 33.3%, with a median TTR of 2 months and median DoR of 7.7 months. As for survival endpoints, the median PFS reached 5.5 months and median OS reached 13 months [56].

As for the HR-positive HER2-negative cohort (N = 54), the ORR was 31.5%, with a median DOR of 8.7 months, median PFS of 5.5 months and median OS of 12 months, at a median follow-up of 11.5 months [89]. Although preliminary, the activity displayed is significant in comparison with other standard options available for heavily pretreated HR-positive metastatic BC [9]. Further evaluation of HR-positive/HER2-negative BC is ongoing in a randomized phase III trial (TROPiCS-02, NCT03901339).

Overall, the toxicity profile was acceptable with nausea, diarrhea, fatigue, anemia and neutropenia as the most common AEs [56]. The most frequent grade ≥3 AEs were neutropenia (50%), anemia (11%) and diarrhea (7.4%). Coherently, treatment discontinuation was caused mostly by neutropenia [56].

The ASCENT study, a randomized phase III trial, investigated the efficacy of SG versus TPC (capecitabine, eribulin, gemcitabine or vinorelbine) in patients with metastatic TNBC who received two or more prior chemotherapies in an advanced setting (N = 468) [90]. SG outperformed TPC in terms of both PFS (median PFS of 5.6 vs. 1.7 months) and OS (median OS 12.1 vs. 6.7 months). Based on these data, in April 2021 the FDA granted regular approval to SG, as treatment for patients with unresectable locally advanced or metastatic TNBC who have received two or more prior systemic therapies, at least one of them for metastatic disease [91].

In the ASCENT trial, the PFS of people with brain metastases seems to favor SG over TPC, according to a subgroup analysis (2.8 vs. 1.6 months) [92]. In this regard, the pharmacokinetic profile of SG in patients with BC brain metastases or primary tumors who would undergo brain surgery has been investigated [93]. SG was administered 16 hours before surgery, with SN-38 and its metabolites investigated through samples from cerebrospinal fluids, intracranial tumor tissue and serum. Patients with BC brain metastases showed therapeutic relevant concentrations of SN-38, with early intracranial responses [93]. In this perspective, a phase II study (SWOG-S2007, NCT04647916) is currently recruiting patients with HER2-negative BC with brain metastases to test the intracranial ORR of SG in this setting [47,93].

Although SG targets Trop-2, the clinical benefit of the ADC versus TPC seems irrespective of the level of Trop-2 expression [90]. However, improved efficacy was observed in Trop-2-high and Trop-2-median BC patients treated with the experimental compound in comparison with TPC [94]. Of note, patients enrolled in ASCENT trial were not stratified according to Trop-2 expression, so no data are available about the efficacy of SG in this setting [9,90].

Consistent with prior studies, the most prevalent grade ≥3 treatment-related AEs were neutropenia (51%), leukopenia (10%), diarrhea (10%) and febrile neutropenia (6%), without fatal events related to the experimental drug (Figure 4) [47].

Finally, a single-arm phase II study in the early setting for TNBC is investigating the pCR rate induced by SG either as monotherapy or in combination with pembrolizumab in the neoadjuvant setting (NeoSTAR, NCT04230109); a randomized phase III study is currently elucidating the efficacy of adjuvant SG versus TPC for residual disease in the post-neoadjuvant setting (SASCIA, NCT04595565) (Table 2).

To date, combinations of SG with other agents are of interest and are being evaluated in the metastatic setting for HER2-negative BC, including ICIs (NCT04468061; NCT04448886; NCT03424005) and poly (ADP-ribose) polymerase (PARP) inhibitors (SEASTAR, NCT03992131; NCT04039230) [47].

#### 4.1.2. Datopotamab Deruxtecan (Dato-DXd, DS-1062)

Dato-DXd is an ADC that is constituted of humanized Trop-2-directed mAb, a tetrapeptide-based linker and a topoisomerase 1 inhibitor payload (exatecan derivative) in early phases of development. The open-label first-in-human phase I study TROPION-PanTumor01 (NCT03401385) clinical trial investigated datopotamab deruxtecan, in patients with advanced solid tumors, including metastatic TNBC. The safety profile is characterized by grade ≥3 AEs that included fatigue, mucosal inflammation such as stomatitis, decreased appetite, nausea, vomiting, constipation, infusion-related reactions, anemia and cough. Notably, 14 patients in the lung cancer cohort experienced ILD, with three grade 5 events [95].

### 4.2. LIV1

LIV-1 is in the family of transmembrane zinc transporter proteins, belonging to the ZIP superfamily [96]. In normal tissues, LIV-1 family expression is heterogeneous [97]. Among this protein family, the LIV1 (ZIP6) protein is typically found in hormonally regulated tissues, such as breast, where its expression seems to be sensitive to estrogen levels [97]. Indeed, *LIV1* was firstly identified as an estrogen-induced gene in BC cell lines; then, it was associated to node involvement in HR-positive BC [97,98]. In addition to BC, LIV1 has been detected in cervical and uterine neoplasms, prostate and pancreatic cancers, as well as in melanoma [9,96,99].

#### Ladiratuzumab Vedotin (SGN-LIV1A)

An anti-LIV1 humanized mAb and a MMAE warhead coupled by means of a cleavable linker constitute the ADC ladiratuzumab vedotin. This compound binds to the extracellular domain of LIV1 and, after internalization, is trafficked to lysosomes where the cytotoxic payload is released by proteolysis [99]. Cancer cells apoptosis is achieved via inhibition of microtubulin polymerization [99]. In TNBC, SGN-LIV1A may induce an effective immunogenic cell death (ICD), potentially improving the benefit from immunotherapy, according to preclinical models [9,100].

SGN-LIV1A is currently investigated in a phase I clinical trial for patients with LIV1-positive metastatic HR-positive/HER2-negative and triple-negative BC (NCT01969643) (Table 2) [101]. At the first data collection, the ORR was 32%, with a median PFS of 11.3 weeks in patients with TNBC treated in the combined dose-escalation and expansion cohorts (N = 44) [47].

In terms of safety, the most common all-grade AEs were fatigue (59%), nausea (51%), peripheral neuropathy (44%), alopecia (36%), decreased appetite (33%), constipation (30%), neutropenia (25%), diarrhea (25%) and abdominal pain (25%) [9,101]. As for grade ≥3 AEs, the most frequent were represented by neutropenia (25%) and anemia (15%) [101].

In early-stage BC, ladiratuzumab vedotin was one of the neoadjuvant treatments planned in the I-SPY2 trial (NCT01042379) and it was administered every 3 weeks for four cycles before doxorubicin/cyclophosphamide (AC) every 2–3 weeks for four cycles [47,102]. Unfortunately, the experimental drug did not improve pCR rates compared to the control arm [102].

The combination of SGN-LIV1A and ICIs has been explored, with two ongoing studies: a combination treatment of ADC plus pembrolizumab as first-line treatment for metastatic TNBC (SGNLVA-002, KY-721, NCT03310957) and ADC plus atezolizumab as second-line treatment (one arm of the Morpheus-TNBC, NCT03424005) [47,103]. In KY-721, among patients who were assessed for efficacy, the ORR was 54% (N = 26). The toxicity profile was manageable, with the most common grade ≥3 AEs represented by neutropenia (8%), diarrhea (8%), fatigue (8%), hypokalemia (8%) and maculo-papular rash (8%).

### 4.3. HER3 (ErbB3)

HER3 is a member of the HER family characterized by weak tyrosine kinase activity. In order to transduce signals downstream, HER3 has to heterodimerize. In this context, HER2 is the most important partner for dimerization [104]. Other high affinity ligands of HER3 are represented by neuregulins (NRG-1 and NRG-2) [105]. A wide variety of cancer histologies overexpress HER3, such as head and neck carcinoma, colorectal cancer, bladder, melanoma, lung, ovarian, prostate and breast cancer [106]. HER3 is believed to be involved in resistance to targeted therapies, not only those against other receptors of the HER family, but also hormonal agents and PI3K-inhibitors [9]. Finally, some oncogenic potential has also been shown by *ERBB3* somatic mutations [107].

#### Patritumab Deruxtecan (U3-1402, HER3-DXd)

Patritumab deruxtecan is a novel anti-HER3 ADC that is composed by the humanized mAb patritumumab and deruxtecan. The mAb is linked to the payload via a peptide-based cleavable linker, with a DAR of 8 [9]. The additional membrane permeability accounts for a potent bystander effect. This experimental compound has been investigated in a phase I/II clinical trial enrolling heavily pretreated patients with HER3-positive metastatic BC, with promising results (NCT02980341) (Table 2) [108]. Patients harboring HER3-high/HR-positive/HER2-negative neoplasms were enrolled into two cohorts to receive the drug at a dose of 4.8 mg/kg or 6.4 mg/kg. In contrast, HER3-low/HR-positive/HER2-negative metastatic BC patients as well as HER3-high metastatic TNBC patients received 6.4 mg/kg of the ADC. At the data cutoff, drug activity was evaluable in 64 patients with HER3-high/HR-positive/HER2-negative metastatic BC. In this group, the ORR was 30% and 13% for patients treated with 6.4 and 4.8 mg/kg, respectively. Among the 31 patients with HER3-low/HR-positive/HER2-negative metastatic BC and the 31 patients with HER3-high metastatic TNBC, the ORR was 33 and 16%, respectively (6.4 mg/kg) [47].

The most common all-grade AEs were gastrointestinal (nausea, 85.7%; appetite reduction, 66.7%; vomiting, 54.8%; increased AST/ALT, 47.6% and 45.2%, respectively) and hematological toxicities (thrombocytopenia, 71.4%; neutropenia, 64.3%; leukopenia, 59.5%; anemia, 38.1%) [9]. Grade ≥3 AEs (≥15%) included thrombocytopenia (35.7%), neutropenia (28.6%), leukopenia (21.4%) and anemia (16.7%), regardless of causality [108].

## 5. Conclusions and Future Perspectives

The emergence of new ADCs with solid efficacy data represents an important therapeutic breakthrough in oncology, particularly in the field of breast cancers (Figure 1).

The progress in ADCs engineering and technology platforms has unlocked the production of new payloads and novel linkers, thus allowing for a new generation of ADCs with high DAR and strong bystander effects. Indeed, membrane-permeable payloads along with new cleavable linkers amplified the effectiveness of the bystander effect, thus potentially extending efficacy to heterogeneous tumors or cancers with homogeneous but low target expression [3]. However, caution is warranted to move these drugs to the early-stage setting, because of the potential risk of serious AEs, including ILD for trastuzumab deruxtecan and neutropenia or diarrhea for SG (Figure 3).

As for future perspectives, new potential targets, such as proteins expressed in the tumor microenvironment or by cancer stem cells, are under evaluation [9]. Additionally, smarter vehicles for payloads are being investigated [9]. In this regard, probody drug conjugates stand out as a new class of recombinant ADCs prodrugs [9]. They can circulate in an inactivated form and are typically activated by proteases through proteolytic cleavage [9]. This optimization of the payload delivery at the tumor site is thought to reduce on-target/off-tumor toxicity [9]. Delivery systems alternative to mAb scaffolds are also in the spotlight. For example, centyrins, small cysteine-free scaffolds, display excellent biophysical properties [9,109]. They can be efficiently internalized by cancer cells and they permit conjugation at various positions [9,109].

Bispecific mAbs and related subsets, such as biparatopic mAbs, are also under evaluation in preclinical models and early-phase studies [9]. Such mAbs are able to simultaneously bind to two different antigens and have demonstrated to improve receptor internalization, lysosomal trafficking and receptor downregulation [9].

Moreover, new payloads, other than cytotoxic agents, are being explored. For example, LMB-100 and ABBV-155, two ADCs with proapoptotic warheads (pseudomonas exotoxin A and a B-cell lymphoma 2, Bcl-2, inhibitor, respectively), are under investigation [9]. Finally, some new ADCs can carry immunomodulatory compounds, such as Toll-like receptor (TLR)7/8 [11].

It has become clear that the era of ADCs strongly relies on ideal patient selection for clinical trial enrollment. In this regard, while some clinical trials focus on specific targets selection by adopting a prescreening phase, others adopt stringent inclusion criteria in order to enroll tumor types with a well-known high target expression, without further individual molecular testing [9].

Such expedients can be problematic, mainly due to the risk of exposing target-negative patients, for whom no clinical benefit is expected, to avoidable toxicity. Conversely, ADCs akin to T-DXd display activity in patients with low target expression as well [3]. The landscape of ADCs clinical development is further complicated by the common lack of validated predictive biomarkers, assays and cutoffs to define antigen status [3,9].

Overall, these aspects need to be carefully studied when designing trials investigating ADCs. Considering that several novel ADCs are in the final steps of clinical development, as phase III clinical trials are ongoing, we will possibly witness a switch from standard treatments, currently based on systemic chemotherapy, to targeted anticancer treatments based on ADCs, either as monotherapy or in combination with other agents [47]. Nevertheless, ADCs development still faces “old” challenges, such as patient selection and biomarker assessment. In future studies, tackling all these aspects to best exploit this class of drugs is warranted.

## Figures and Tables

**Figure 1 cancers-13-02898-f001:**
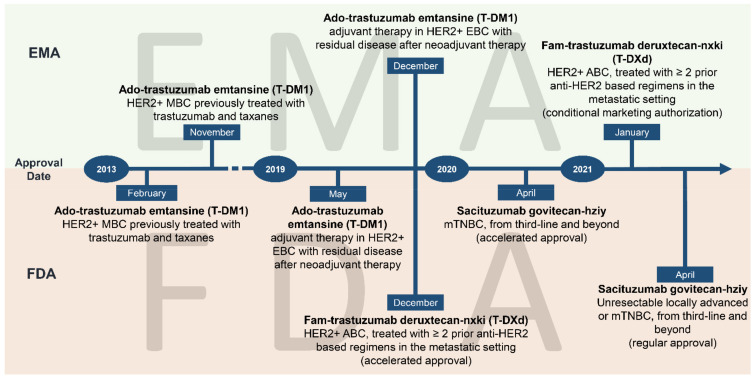
Temporal milestones of antibody–drug conjugates currently approved for the treatment of BC. Abbreviations: HER2, human epidermal growth factor receptor 2; T-DM1, trastuzumab emtansine; T-Dxd, trastuzumab deruxtecan; EMA, European Medicines Agency; FDA, Food and Drug Administration; 1MBC, metastatic breast cancer; 2EBC, early breast cancer; 3ABC, advanced breast cancer; mTNBC, metastatic triple-negative breast cancer.

**Figure 2 cancers-13-02898-f002:**
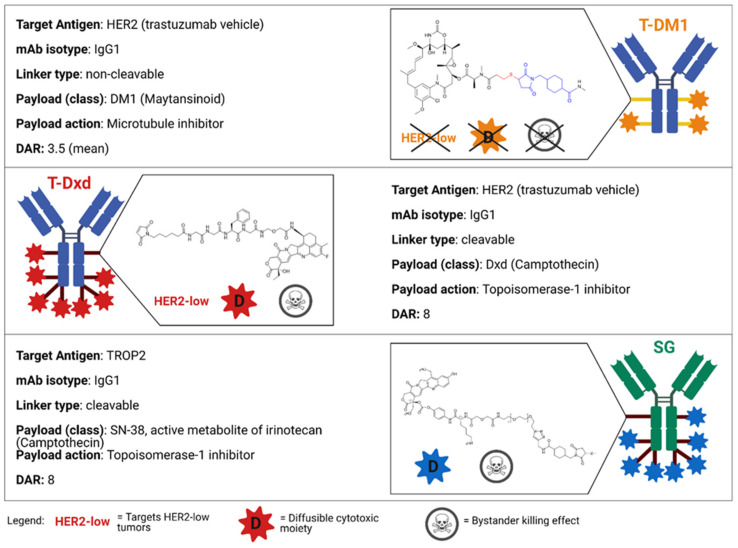
Main features of ADCs currently FDA-approved for the treatment of BC. Abbreviations: T-DM1, trastuzumab emtansine; T-DXd, trastuzumab deruxtecan; SG, Sacituzumab govitecan; HER2, human epidermal growth factor receptor 2; mAb, monoclonal antibody; DAR, drug-to-antibody ratio; DM1, emtansine; DxD, deruxtecan; TROP2, Trophoblast cell surface antigen 2; TNBC, triple-negative breast cancer. Created with Biorender.com (accessed on 15 March 2021).

**Figure 3 cancers-13-02898-f003:**
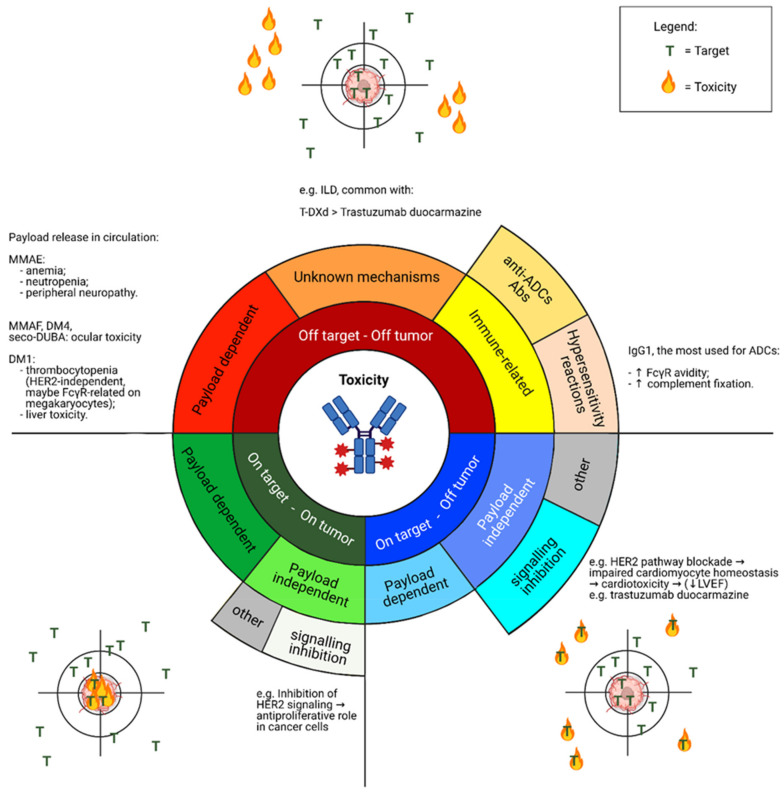
Spectrum of toxicity observed in clinical trials investigating novel ADCs. Abbreviations: ILD, interstitial lung disease; FcγR, Fragment crystallizable-gamma receptor; T-DXd, trastuzumab deruxtecan; MMAE, Monomethyl auristatin E; MMAF, Monomethyl auristatin F; DM1, mertansine/emtansine; DM4, ravtansine/soravtansine; seco-DUBA, seco-duocarmycin-hydroxybenzamide-azaindole; LVEF, Left Ventricular Ejection Fraction; Abs, antibodies. Created with Biorender.com (accessed on 15 March 2021).

**Table 1 cancers-13-02898-t001:** Summary of current clinical trials investigating antibody drug conjugates targeting HER2. Legend: ° Active, not recruiting; * Not yet recruiting. Abbreviations: ID, identifier; #, number; pts, patients; BC, breast cancer; mBC, metastatic breast cancer; HER2, human epidermal growth factor receptor 2; CT, chemotherapy; ET, endocrine therapy; HR, hormone receptor; TPC, treatment of physician’s choice; NSCLC, non-small cell lung cancer; mTNBC, metastatic triple negative breast cancer; eBC, early breast cancer; T-DM1, trastuzumab emtansine; PD-L1, programmed death-ligand 1. Source: ClinicalTrial.gov (accessed on 15 March 2021).

Drug	Trial ID	Ph	# of Pts	Patient Cohort	Treatment Arms
ALT-P7	NCT03281824 °	I	30	HER2-positive mBC pretreated with trastuzumab	ALT-P7
ARX788	NCT03255070	I	60	HER2-positive and HER2-low mBC; HER2-positive gastric cancer	ARX788
A166	NCT03602079	I/II	82	HER2-expressing (≥1+) solid tumors (including BC)	A166
BAT8001	NCT04185649 °	III	410	HER2-positive mBC pretreated with trastuzumab and a taxane	BAT8001 vs. Capecitabine + Lapatinib
DHES0815A (RG6148)	NCT03451162 °	I	14	HER2-positive mBC	DHES0815A
Disitamab Vedotin (RC-48)	NCT04280341 *	I	50	Advanced HER2-positive solid tumors (including BC); standard treatment-refractory	RC-48 + JS001 (anti-PD1)
	NCT03052634	I/II	111	HER2-positive and HER2-low mBC for which no standard therapy exists	RC-48
	NCT03500380	II	228	HER2-positive mBC. Prior treatment with trastuzumab and taxane; up to two prior CT in advanced setting	RC-48 vs. Capecitabine + Lapatinib
	NCT04400695	III	366	HER2-low mBC. Pretreated with one or two prior CT (anthracyclines required) and ET if HR+	RC-48 vs. TPC
FS-1502	NCT03944499 *	I	92	HER2-expressing advanced solid tumors and HER2-positive mBC	FS-1502
GQ1001	NCT04450732	I	27	HER2-positive advanced solid tumors (including BC)	GQ1001
PF-06804103	NCT03284723 °	I	148	HER2-positive and HER-2 negative mBC	PF-06804103 +/− (Palbociclib +Letrozole)
Trastuzumab Deruxtecan (DS-8201a)	NCT03523572 °	I	99	HER2-positive and HER2-low mBC or urothelial cancer	T-DXd + Nivolumab
	NCT04556773 (DESTINY-Breast08)	I	185	HER2-low mBC	T-DXd + Capecitabine T-DXd + Durvalumab and Paclitaxel T-DXd + Capivasertib T-DXd + Anastrazole T-DXd + Fulvestrant
	NCT04042701	I	115	HER2-positive and HER2-low mBC or HER-2-expressing/mutant NSCLC	T-DXd + Pembrolizumab
	NCT04538742 (DESTINY-Breast07)	I/II	350	HER2-positive mBC. At least one prior treatment line in metastatic setting required (part 1); no prior lines of therapy for advanced/mBC allowed (part 2).	T-DXd T-DXd + Durvalumab T-DXd + Pertuzumab T-DXd + Paclitaxel T-DXd + Durvalumab and paclitaxel T-DXd + Tucatinib
	NCT03742102 (BEGONIA)	I/II	200	First-line treatment in mTNBC (Arm 6 HER2-low)	Durvalumab + other agents (T-DXd in Arm 6)
	NCT04539938 (HER2CLIMB-04)	II	70	HER2-positive mBC treated with at least two prior anti-HER2-based regimens in the metastatic setting	T-DXd + Tucatinib
	NCT04752059 (TUXEDO-1)	II	15	HER2-positive BC with brain metastases and prior exposure to trastuzumab and pertuzumab.	T-DXd
	NCT04553770	II	88	HR+/HER2-low eBC candidate to neoadjuvant therapy	T-DXd +/− Anastrazole
	NCT04420598 (DEBBRAH)	II	39	HER2-positive or HER2-low BC with brain or leptomeningeal metastases	T-DXd
	NCT04622319 (DESTINY-Breast05)	III	1600	HER2-positive eBC with residual invasive disease following neoadjuvant therapy containing trastuzumab and taxane	T-DXd vs. T-DM1
	NCT04494425 (DESTINY-Breast06)	III	850	HR+/HER2-low mBC. Progression on at least two previous lines of ET; no prior CT for advanced or mBC	T-DXd vs. TPC
	NCT03529110 (DESTINY-Breast03) °	III	500	HER2-positive mBC previously treated with trastuzumab and a taxane	T-DXd vs. T-DM1
	NCT03734029 (DESTINY-Breast04) °	III	557	HER2-low mBC. If HR+ has progressed on ET; one to two prior CT for mBC	T-DXd vs. TPC
	NCT03523585 (DESTINY-Breast02) °	III	600	HER2-positive mBC previously treated with T-DM1	T-DXd vs. TPC
	NCT04739761 * (DESTINY-Breast12)	IV	500	HER2-positive mBC	T-DXd
Trastuzumab Duocarmazine (SYD985)	NCT04602117 *	I	27	HER2-positive and HER2-low pretreated solid tumors (only BC in expansion cohort)	SYD985 + Paclitaxel
	NCT04235101	I	120	HER2-expressing (≥1+) solid tumors (including BC) for which no standard therapy exists	SYD985 + Niraparib
	NCT03262935 ° TULIP	III	436	HER2-positive mBC progressed after at least two prior anti-HER2-based regimens or after T-DM1 for advanced or mBC	SYD985 vs. TPC
Trastuzumab Emtansine (T-DM1)	NCT01976169 °	I	28	HER2-positive and RB-proficient mBC previously treated with anti-HER2	T-DM1 + Palbociclib
	NCT03032107 °	I	27	HER2-positive mBC; prior treatment with trastuzumab and a taxane	T-DM1 + Pembrolizumab
	NCT04509596	I	94	HER2-positive mBC which failed prior therapies	DZD1516 +/− Capecitabine and Trastuzumab or +/− T-DM1
	NCT02236000 °	I/II	63	HER2-positive mBC pretreated with trastuzumab and pertuzumab	T-DM1 + Neratinib
	NCT02657343 °	I/II	26	HER2-positive mBC. In Cohort A: prior trastuzumab and taxane, up to four prior lines in the metastatic setting	Ribociclib + other agents (Cohort A with T-DM1)
	NCT03190967	I/II	125	HER2-positive BC with treated brain metastases	T-DM1 +/− Metronomic Temozolomide
	NCT03587740 ° (ATOP TRIAL)	II	82	HER2-positive mBC in older patient (≥60 years)	T-DM1
	NCT01853748 ° (ATEMPT Trial)	II	512	Stage I HER2-positive BC, adjuvant treatment	T-DM1 vs. Paclitaxel/Trastuzumab
	NCT04733118 * (PHERGAIN-2)	II	393	HER2-positive untreated eBC	T-DM1 + Pertuzumab + Trastuzumab
	NCT03530696	II	132	HER2-positive mBC. Prior treatment with pertuzumab, no more than two lines of therapy for mBC	T-DM1 +/− Palbociclib
	NCT04158947 (HER2BAT)	II	130	HER2-positive BC with central nervous system recurrence/progression during or after an anti-HER2 therapy	T-DM1 +/− Afatinib
	NCT04351230	II	126	HER2-positive mBC progressed on treatment with a taxane, trastuzumab and pertuzumab	T-DM1 +/− Abemaciclib
	NCT03084939 °	III	351	Chinese patients with HER2-positive mBC who have received prior taxane and trastuzumab.	T-DM1 vs. Capecitabine + Lapatinib
	NCT04740918 * (KATE3)	III	350	HER2-positive and PD-L1 positive mBC pretreated with trastuzumab and taxane	T-DM1+/− Atezolizumab
	NCT03975647 (HER2CLIMB-02)	III	460	HER2- positive mBC; prior treatment with taxane and trastuzumab	T-DM1 +/− Tucatinib
	NCT04457596 (CompassHER2 RDTrial)	III	1031	High risk HER2-positive BC with residual disease after neoadjuvant therapy	T-DM1 +/− Tucatinib

**Table 2 cancers-13-02898-t002:** Summary of the current clinical trials investigating antibody drug conjugates with targets other than HER2. Legend: ° Active, not recruiting; * Not yet recruiting. Abbreviations: ID, identifier; pts, patients; BC, breast cancer; EGFR, epidermal growth factor receptor; mTNBC, metastatic triple negative breast cancer; mBC, metastatic breast cancer; HER2, human epidermal growth factor receptor 2; CT, chemotherapy; ET, endocrine therapy; HR, hormone receptor; TPC, treatment of physician’s choice; eBC, early breast cancer; FRA, folate receptor alpha; HER3, human epidermal growth factor receptor-3; PD-L1, programmed death-ligand 1; 5T4, oncofetal antigen 5T4; ROR2, receptor tyrosine kinase-like orphan receptor 2; TROP-2, Trophoblast cell surface antigen 2; LIV-1, Zinc transporter ZIP6; FRA, Folate receptor alpha; ROR1, receptor tyrosine kinase-like orphan receptor 2; HER3, human epidermal growth factor receptor 3; BRCA1, Breast Related Cancer Antigen 1; BRCA2, Breast Related Cancer Antigen 2; PALB2, Partner and localizer of BRCA2; RAD51C, RAD51 homolog C; RAD51D, RAD51 homolog D; CPS, Combined Positive Score. Source: ClinicalTrial.gov, accessed on 15 March 2021.

Drug	Target	Trial ID	Study Ph	# of Pts	Patient Cohort	Treatment Arms
ASN004	5T4	NCT04410224 *	I	43	Advanced solid tumors (including BC) with literature evidence of 5T4 expression	ASN004
AVID100	EGFR	NCT03094169 °	I/II	90	Advanced EGFR-overexpressing solid tumors (including TNBC)	AVID100
BA3021 (CAB-ROR2-ADC)	ROR2	NCT03504488	I/II	120	Advanced solid tumors (including TNBC)	CAB-ROR2-ADC
Datopotamab Deruxtecan (DS-1062)	TROP-2	NCT03401385 (TROPION-PanTumors-01)	I	770	Advanced solid tumors (including TNBC and HR+/HER2-negative BC)	DS-1062
		NCT03742102 (BEGONIA)	I/II	200	First-line treatment in mTNBC	Durvalumb + other agents (DS-1062 in Arm 7)
Enfortumab Vedotin (ASG-22ME)	NECTIN-4	NCT04225117 (EV-202)	II	240	Advanced solid tumors (including HR+/HER2-negative and TNBC)	Enfortumab Vedotin
Ladiratuzumab Vedotin (SGN- LIV1a)	LIV-1	NCT01969643	I	418	Pretreated mBC (all subtypes)	SGN- LIV1a +/− Trastuzumab (if HER2-positive)
		NCT03310957	I/II	122	mTNBC; first-line treatment	SGN-LIV1A + Pembrolizumab
		NCT03424005 (Morpheus-TNBC)	I/II	280	mTNBC	Umbrella study including a combination of SGN-LIV1A plus Atezolizumab
MEN1309 (OBT076)	CD205	NCT04064359	I	70	CD205-positive HER2-negative advanced solid tumors (including BC)	OBT076
MORAb-202	FRA	NCT04300556	I/II	196	FRA-positive TNBC, NSCLC, endometrial and ovarian cancer	MORAb-202
NBE-002	ROR1	NCT04441099	I/II	100	Advanced solid tumors (including TNBC)	NBE-002
Patritumab Deruxtecan (U3-1402)	HER3	NCT04610528 (TOT-HER3)	I	80	HR+/HER2-negative eBC	U3-1402
		NCT02980341 °	I/II	180	HER3-positive mBC: no standard treatment available	U3-1402
		NCT04699630 *	II	120	mBC. For TNBC: at least one but no more than three prior lines of CT. For HR+: no limit to prior ET but no more than two prior CT.	U3-1402
Sacituzumab Govitecan (IMMU-132)	TROP-2	NCT04617522 *	I	24	Advanced solid tumor (including BC) in patients with moderate liver impairment	IMMU-132
		NCT04039230	I/II	75	mTNBC	IMMU-132 + Talazoparib
		NCT03424005 (Morpheus-TNBC)	I/II	280	mTNBC	Umbrella study including a combination of IMMU-132 plus atezolizumab
		NCT03992131 (SEASTAR)	I/II	329	Advanced solid tumor (including TNBC) with a deleterious mutation in BRCA1, BRCA2, PALB2, RAD51C or RAD51D	Rucaparib + IMMU-132 or plus Lucitanib
		NCT04230109 (NeoSTAR)	II	100	Localized TNBC candidate for neoadjuvant therapy	IMMU-132 +/− Pembrolizumab
		NCT04647916	II	44	HER2-negative BC with brain metastases	IMMU-132
		NCT04468061 (Saci-IO TNBC)	II	110	Treatment-naïve mTNBC; PD-L1 negative	IMMU-132 +/− Pembrolizumab
		NCT04448886 (Saci-IO HR+)	II	110	HR+/HER2-negative mBC; PD-L1 positive (CPS ≥10). Progression on or within 12 months of adjuvant ET or on at least one line of ET for metastatic disease	IMMU-132 +/− Pembrolizumab
		NCT04454437 °	II	80	Chinese patients with mTNBC refractory to at least two lines of CT for mBC	IMMU-132
		NCT04595565 (SASCIA)	III	1200	HER2-negative BC with residual disease after neoadjuvant chemotherapy	IMMU-132 vs. TPC
		NCT03901339 (TROPICS-02)	III	400	HR+/HER2-negative mBC who have failed at least two prior CT regimens	IMMU-132 vs. TPC
		NCT04639986	III	330	HR+/HER2-negative mBC after failure of at least 2, and no more than 4, prior CT for metastatic disease	IMMU-132 vs. TPC

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
