# Peer review of "Antibody–Drug Conjugates for the Treatment of Breast Cancer"

_cancers, 2021, doi:10.3390/cancers13122898_

Round 1

Reviewer 1 Report

The review article entitled “Antibody-Drug Conjugates for the Treatment of Breast Cancer “ by  Corti C. et al. shows a global point of view of the development of Antibody-drug conjugates in the treatment of breast cancer by discussing the limitations of this innovative therapeutic approach. This is a nice manuscript, well-written, informative, and broken down into sections that provide the readers with a logical flow of the content. The figures included in the manuscript are accurate. Further, the paper has a good and balanced choice of the reviewed papers.

Author Response

We are particularly grateful to Reviewer #1  for appreciating our work.

Reviewer 2 Report

I think this paper is very well-organized and presented and refers to a hot topic

Author Response

We are particularly grateful to reviewer for appreciating our work. 

Reviewer 3 Report

Corti et al. provide a well organiseds and easy to follow review about the current antibody-drug conjugates for the treatment of Breast Cancer. They mainly focus on drugs currently found in clinical trials. 

See comments below:

Comment #1 - It results difficult to understand the overall survival and progression-free disease data, as no reference point it is indicated. For example, in line 270-271: " "with an ORR of 58.3% and a median PFS of 18.1 months" --> how good or bad is 58.3% and 18.1 months? In line 475-476 the authors use a more compressible way to provide ORR and PFS data, i.e. "indicating for example 2.8 vs 1.6 months", as they provide a reference point. 

Comment #2 - I was missing some information about ADC that are currently being developed. Personally, I would like to have more information about new strategies than about the adverse effects of all these ADC. In the same context, the authors state in the abstract that in this review they will discuss the current landscape about ADCs development.... but in the review there is minor information about what is being developed... 

Comment #3 - some abbreviations are not well defined, or they are defined later. for example T-DM1 is first mentioned in line 104 and described in line 188. pCR is first mentioned in line 218 and defined in line 232. 

Comment #4 - the mechanism of action of defined in only few payload, but not all.  

Author Response

We are grateful to the editor and to the reviewers for carefully reading our manuscript and for kindly providing us with their suggestions.

Reviewer #3

  • Comment #1: we thank Reviewer #3 for the comment. We provided in the text a reference to the comparison with the general study population. Since DESTINY-Breast01 trial was a single-group, open-label, phase 2 trial of T-DXd a mirror comparator for the subgroup with brain metastases is not available. However, we believe it is still useful to provide such data, considering the particular poor prognosis of this breast cancer pts subgroup and the general lack of evidence for this setting.
  • Comment #2: we thank Reviewer #3 for the comment. A lot of the trials discussed in the review are early-phase trials about ADC not yet used in daily clinical practice. In this sense we are providing a current landscape of ADC development. Additionally, in the final part of the manuscript the authors provide possible new strategies that will possibly be soon investigated in the early-phase.
  • Comment #3: we are grateful to Reviewer #3 for the comment. T-DM1 is first mentioned in line 58, and not in line 104 as Reviewer 3 wrote. The abbreviation for pCR has been amended.